# Potential Regulatory Gene Network Associated with the Ameliorative Effect of Oat Antibacterial Peptides on Rat Colitis

**DOI:** 10.3390/foods13020236

**Published:** 2024-01-11

**Authors:** Helin Wang, Xiaoxing Chi, Dongjie Zhang

**Affiliations:** 1College of Food Science, Hei Long Jiang Bayi Agricultrue University, Daqing 163319, China; 18245705737@163.com (H.W.); liuchangzan@126.com (D.Z.); 2Key Laboratory of Agro-Products Processing and Quality Safety of Heilongjiang Province, Daqing 163319, China; 3Coarse Cereals Engineering Research Center, Heilongjiang Bayi Agricultural University, Daqing 163319, China

**Keywords:** oat protein, antimicrobial peptide, anti-inflammatory, DAG, T cell, regulatory network

## Abstract

Oat protein is unstable in intestinal fluid digestion, and it is easily degraded by trypsin and chymotrypsin, producing low molecular weight peptides. Endopeptidase hydrolysis can improve the bioavailability of active peptides and avoid further digestion in the gastrointestinal tract. Antimicrobial peptides (AMPs) can effectively improve host immunity, but most related studies focus on physiology and ecology, and there are few reports on their molecular level. Therefore, in this article, oat peptides were prepared via the simulated digestion method in vitro, and the main metabolites and action factors affecting colitis were screened by using the multi-omics methods in a high-throughput mode to analyze the effect and mechanism of colitis. Firstly, oat antimicrobial peptides were prepared from cationic resin combined with HPLC, and the anti-inflammatory effects of antimicrobial peptides were analyzed in vitro through the use of human colon epithelial (HCoEpiC) anti-inflammatory cells. In vivo experiments using rats have verified that AMPs can effectively prevent colitis caused by dextran sodium sulfate (DSS), reduce intestinal inflammatory cell infiltration and glandular disappearance in the colon, and reduce the apoptosis rate of colon cells. Secondly, metabolomics and transcriptomics were combined to analyze the mechanism of preventing enteritis, and it was found that oat antimicrobial peptides can promote DAG diglycerol production and inhibit the activation of T helper cells (TH), resulting in the down-regulation of key factors in the main downstream pathways of TH1, TH2 and TH17, and inhibit the production of inflammatory cells. At the same time, AMP can activate the wnt pathway, improve the expression of key genes of wnt and frizzled, promote the generation of intestinal stem cells, facilitate the differentiation and repair of intestinal epithelial cells, and prevent the generation of enteritis. Finally, the underlying genetic regulatory network of the important pathway was constructed from the effect of AMP on rat colitis.

## 1. Introduction

Inflammatory bowel disease (IBD) is a typical chronic intestinal inflammatory disease. Although the therapeutic effect is mainly achieved by regulating or inhibiting the excessive inflammation with drugs, there are visible side effects [1]. Antibacterial peptides are natural bioactive molecules that can not only kill bacteria directly but also indirectly via their immuno-regulatory effects, including inducing cytokine secretion and through the recruitment of macrophages, inducing the synthesis of pro-inflammatory factors, reducing inflammation caused by endotoxins [2,3]. Furthermore, they can regulate the tendency of immune cells and the differentiation of lymphocytes and promote the production of lymphocytes, cytokines and tendency factors [4,5]. Antimicrobial peptide LL-37 can regulate the apoptosis and activation of macrophages and neutral cells [6], promote the clearance of host pathogens and enhance anti-infection ability. Chronic inflammation leads to continuous bacterial stimulation of the host, resulting in tissue and organ damage. Antimicrobial peptides have antibacterial and host immune regulation effects, and antimicrobial peptides have gradually attracted attention in relation to the treatment of chronic inflammation [7,8].

Current relevant data show that colitis mainly damages intestinal mucosal epithelial cells, thereby promoting the release of inflammatory mediators, including IL-4, IL-6, IL-10, IL-12, IL-1p and TNF-α, which bind to the surface receptors of intestinal mucosal epithelial cells. This activates the expression of related pathways or genes, including Lgr5, Ascl2, OCT4, Olfm4, etc. [9,10]. It also promotes the enlargement of the stem cell pool, accelerates the migration and differentiation of stem cells, and promotes the repair of the injured mucosal epithelium [11,12]. Lgr5 was considered to be an ISC marker [13].

At present, genomic information concerning the rat colon provides an important premise and reference basis for high-throughput technologies, such as transcriptomics, proteomics and metabolomics. Therefore, in this study, the colon tissue structure, cell morphology, metabonomics and transcriptome were comprehensively analyzed to identify the molecular mechanism of the effect of oat antimicrobial peptides on colitis. Here, five animal model groups (normal group, colitis group, low-dose antimicrobial peptide group, medium-dose antimicrobial peptide group, and high-dose antimicrobial peptide group) were selected for further analysis of colon stem cell synthesis and RNA-Seq data. The comprehensive analysis of RNA-Seq data provided a lot of information on the physiological and biochemical changes of oat antimicrobial peptides occurring during the repair of colitis. Finally, metabonomics and transcriptomes were comprehensively analyzed to reveal the important pathways related to the repair of colitis.

## 2. Main Materials and Methods

### 2.1. Materials

Oat protein collection was undertaken in September 2022 by Sian Biotechnology Co., Ltd., Xi’an, China. Porcine pepsin (30,000 U/g) and Trysin (250 U/g) were purchased from Shanghai Jizhi Biochemical Technology Co., Ltd., Shanghai, China. Cation resin LSI-010, SOD, MDA, GSH-PX, MPO kits were purchased from Nanjing Jian cheng Bioengineering Institute, Nanjing, China. CMCC44102 *E. coli* and CMCC26003 *S. aureus* were obtained from Biao Bowei Biotechnology Co., Ltd., Beijing, China. Human colonic epithelial cells (HCoEpiC) were obtained from Guangzhou Gini Eurobio Technology Co., Ltd., Guangzhou, China. Female SD rats (200–220 g, 6–7 weeks) were purchased from Chang sheng Biotechnology Co., Ltd., Liaoning, China. Transcription and protein group sequencing was undertaken by Beijing Biomarker Technologies Co., Ltd., Beijing, China and enrichment and differentially expressed gene annotation analysis was achieved through the BMK Cloud Analytics platform (https://international.biocloud.net/zh/dashboard, accessed on 12 April 2022).

### 2.2. Preparation of Oat Antimicrobial Peptides

For the simulation of oral digestion, 5 g of oat protein and oral saliva were diluted at a ratio of 1:1 (*w*/*w*). The simulated saliva was composed of 4 mL of 1.25× electrolyte stock, 0.3 M/mL of 0.025 mL CaCl_2_, and 0.975 mL of water at 37 °C with pH 7, which was subsequently stirred for 2 min.

For the simulation of gastric digestion, 10 mL from the oral phase was mixed with gastric digestive fluid at 1:1 (*v*/*v*). The gastric digestive fluid was composed of 8 mL of 1.25× electrolyte stock, 0.3 M/mL of 0.005 mL CaCl_2_, 0.4 mL of 5 mol/L HCL, 0.667 mL of Porcine pepsin (2000 U/mL), and 0.928 mL of water, which was prepared into a 20 mL solution at 37 °C with a pH 3.0, which was subsequently stirred for 2 h, and then the pH was increased to 7, partially inactivating the pepsin.

For the simulation of small intestine digestion, 20 mL from the gastric phase and intestinal digestive fluid were mixed 1:1 (*v*/*v*). The intestinal digestive juices consisted of 8 mL of 1.25× electrolyte stock, 0.3 M/mL of 0.04 mL CaCl_2_, 5 mL trypsin activity (100 U/mL), and 6.16 mL of water, respectively. The solution had a pH of 7.0, was maintained at 37 °C, and then stirred for 2 h. Then, 100 µL of BBI solution (0.05 g/L) was added to 1 mL of intestinal chyme water after completion. This was then centrifuged at 3500 rpm for 10 min, the supernatant was collected, and a freeze-drying machine was used to dry the solution [14].

Oat peptides with a molecular weight of <5000 Da were obtained via an ultrafiltration centrifuge tube, and an LS-010 cationic resin column with a volume of 200 mL in the chromatographic column was used for adsorption. Then, anion and uncharged oat peptides were removed via elution with deionized water, and positively charged oat peptides were obtained via elution with 0.3 mol/L NaOH and 4 mL/min. The light absorption value was determined at 280 nm, and the spectrum was plotted. CMCC44102 *E. coli* and CMCC26003 *S. aureus* were used as indicator bacteria, and the indicator bacteria concentration was 10^6^–10^8^ cfu/mL in the bacteriostatic zone experiment, as previously described [15].

### 2.3. Oat Antimicrobial Peptides Sequence Analysis Based on LC-MS/MS

Oat antimicrobial peptides were reduced with 10 mM DTT and methylated for 40 min, and then dissolved in 2–20 μL 0.1% formic acid for LC-MS/MS analysis. The LC-MS/MS analysis conditions were as follows: the sample loading volume was 5 µL, and the mobile phase separation was 0.1% formic acid aqueous solution (A) and 0.1% formic acid acetonitrile solution (B). A gradient elution was used. The elution time was divided into five periods of 0–5 min, 5–20 min, 20–50 min, 50–58 min and 58–60 min, and the concentration ranges of eluent B were 6–9%, 9–14%, 14–30%, 30–40% and 40–95%, respectively.

The conditions for mass spectrometry analysis were as follows: the spray voltage was 2.2 kV, the capillary temperature was 270 °C, the MS resolution was 400 *m*/*z*, 70,000, and the he *m*/*z* range of the MS precursors was 400~1200.

### 2.4. Determination of Anti-Inflammatory Activity of Antimicrobial Peptides In Vitro

The determination of anti-inflammatory activity in vitro was slightly modified with reference to the experimental method of Yan [16]. The experiment was divided into five groups: the blank (control), negative (model), high dose (AMP-H), middle dose (AMP-M) and low dose (AMP-L) experimental groups.

The experimental groups were added to 50 μg/mL, 100 μg/mL and 200 μg/mL antimicrobial peptides, respectively. The blank group and the control group were not added to the antimicrobial peptides. The inflammatory factors TNF-α, IL-6, IL-1β and anti-inflammatory factors IL-4 and IL-6 in HCoEpiC cells were detected through the use of ELISA. The experiments were operated using the procedure according to the ELISA kit.

### 2.5. DSS-Induced Colitis Model and Preventive Effects of Colitis Assessment

The experiment was divided into five groups: (1) the normal model group that only drank water (BLANK); (2) the negative control group that drank water containing 5% dextran sulfate sodium (DSS) (NEG); (3) the low-dose group that drank water containing 5% DSS and 10 mg AMP/200 g rat body weight (L-AMP); (4) the medium-dose group that drank water containing 5% DSS and 20 mg AMP/200 g rat body weight (M-AMP); and (5) the high-dose group that drank water containing 5% DSS and 40 mg AMP/200 g rat body weight (H-AMP). After 14 days of feeding and fasting for 12 h, the colon, colon mucosa and colon contents of the rats were frozen with dry ice and preserved at −80 °C.

Concerning the colonic tissue structure and cell structure detection, the distal colons were 2 cm in size, as with the experimental sample.

Colonic tissue structure and cell structure were evaluated using the H&E staining method [17], while TUNEL-mediated apoptosis detection was evaluated using the in situ TUNEL apoptosis detection kit (product code JC7022).

For colonic antioxidant capacity detection, 1.000 g of colon tissue was added to 9 mL of cold salt water, ground, centrifuged and the supernatant was removed prior to 20 dilutions, and then the antioxidant activity was determined through the use of kits.

### 2.6. Metabolomics and Transcriptome Combined Analysis

Metabolomics and transcriptomics tests were commissioned by BioMarker Technology Co., Ltd., Beijing, China.

For global and differential gene expression analysis, genes with |log2FC| > 1 and a false discovery rate (FDR) < 0.05 were identified as differentially expressed genes (DEGs). All identified DEGs (FDR ≤ 0.05) were then subjected to KEGG pathway enrichment analysis.

For gene co-expression network analysis (GSEA), the KEGG-enriched pathway was analyzed by using GSEA to determine the DEGs (*p* < 0.05, NES > 1) affecting the pathway.

GSEA analysis demonstrated significantly up-regulated and down-regulated gene clusters in AMP-regulated enteritis-related pathways and identified Hub genes and their relationship networks.

### 2.7. Western Blot Analysis

The key control factors were selected for Western blot to validate the inflammatory factors that could promote pathway metabolism of the wnt/β-catenin metabolic pathway and increase the expression of SCI. First, the total protein was isolated from colon tissue. Second, the total protein was measured by using a BCA kit. The stock protein solution (2 µg/µL) was separated via SDS electrophoresis. Then, 5% milk-TBST or 5% BSA-TBST were used to dilute the first antibody. Then, 8 µL of antibody stock solution was added to 4000 µL antibody diluent. Goat-Anti-Mouse IgG-HRP (1:10,000) was used to dilute the second antibody. Finally, the bands were detected using the luminescent kit (Code C506668-0100).

### 2.8. Statistical Analysis

Each experiment was carried out using three parallel experiments, and the experimental data were analyzed in terms of variance using spss version 15.0 software. Sigma Plot 10.0 was used to plot the graphs.

## 3. Results and Discussion

### 3.1. Enrichment of Antimicrobial Peptides and Determination of Anti-Inflammatory Activity

The experiment simulated in vitro digestion, and oat protein was digested through oral chewing, gastric juice and intestinal juice to obtain oat peptides. According to the mechanism of action of antimicrobial peptides, positively charged peptides would specifically target and act on negatively charged cell walls and membranes to exert antibacterial effects [18]. Therefore, oat protein digestion was purified through the use of cationic resin LSI-010 to obtain positively charged oat peptides, which can effectively inhibit the growth of Escherichia coli and Staphylococcus aureus (Figure 1A,B). When the positive charge number of antimicrobial peptides obtained from cottonseed protein hydrolysate separated from cationic resin increased from 2 to 5, the antibacterial activity increased by 4.8 times. The inhibition zones of 4.0 mg/mL cottonseed protein antimicrobial peptide against Escherichia coli and Staphylococcus aureus were 13 mm and 11 mm, respectively [19]. Capsicum seed protein antimicrobial peptide can significantly inhibit *E. coli* and *S. aureus* [20]. From the above analysis, it can be inferred that plant antimicrobial peptides can effectively inhibit Gram-positive and Gram-negative bacteria.

LPS promoted the production of pro-inflammatory factors TNF-α, IL-6 and IL-1β in HCoEpiC cells, inhibited the production of anti-inflammatory factors IL4 and IL10, destroyed the immune system balance of HCoEpiC cells, and led to cellular inflammation. In vitro experiments showed that 100 μg/mL and 200 μg/mL oat antimicrobial peptides could effectively reduce the expression of pro-inflammatory factors and promote the expression of anti-inflammatory factors in HCoEpiC cells (Figure 1C). These results indicate that oat antimicrobial peptides isolated from cationic resin can improve cellular immune activity and prevent inflammation.

### 3.2. Determination of Primary Structure of Oat Antimicrobial Peptides

Through LC-MS/MS analysis, a total of 863 oat AMPs with molecular weights of <1500 Da were obtained (Figure 2), among which five oat antimicrobial peptides with scores > 500 were obtained with two charges, and the two ends of the peptides were hydrophobic amino acids leucine, valine and isoleucine and positively charged lysine residues (Table 1). Hydrophobic amino acids leucine, valine and isoleucine can interact better with the lipophilic region of the lipid bilayer of the cell membrane, while positively charged amino acids can interact with the negatively charged cell wall and membrane to play an antibacterial role [21,22].

### 3.3. Function Evaluation of AMP for Preventing Colitis in Rats

SOD, MDA, GSH-PX and MPO were determined in the colon of the five groups of animal models. In antioxidant experiments, NEG was significantly different from M-AMP and extremely significantly different from H-AMP. The research showed that SOD and GSH-PX in the experimental group effectively increased, while MDA and MPO decreased, indicating that oat antimicrobial peptides could improve the antioxidant capacity of the colons (Figure 3A). Additionally, through hematoxylin and eosin (H&E) staining (Figure 3B), as shown in the normal rat colon (BLANK), the wall structure was clear, the glands were neatly arranged, and the intestinal mucosal epithelium was continuous. The negative control group (NEG) rat colon’s glandular structure disappeared and was replaced by inflammatory tissues. The rat colon inflammatory cells in the L-AMP group were increased in the colon lamina. However, the colons of the M-AMP and H-AMP rats showed no abnormalities. Meanwhile, the colon positive rate was negatively correlated with the AMP dose (Figure 3C). Finally, the 20 mg/200 g AMP dosage was determined to be effective in preventing colitis.

MPO is an important regulator of inflammation. When MPO is released after body injury, MDA is the main end product of lipid peroxidation and also a marker for the determination of oxidative stress ability of cell components. Rashidian found that 1 mg/mL of antimicrobial peptide CM11 can improve the body’s antioxidant capacity [23]. These results indicated that oat antimicrobial peptides could improve the antioxidant capacity of the colon and effectively prevent colon injury caused by DSS.

### 3.4. Mechanism of Action of AMP to Prevent Colitis

#### 3.4.1. Expression Trend of Metabolites during Anti-Colitis Action of Antimicrobial Peptides

The main difference metabolites of |log2FC| > 1.5, *p*-value < 0.05 and VIP > 1 in AMP and NEG groups were screened out via metabolomics. As can be seen from Table 2, oat antimicrobial peptides can promote the formation of phospholipids PC and PE and reduce the content of amino acids. According to the metabolic pathway diagram shown in Figure 4, oat antimicrobial peptides converted Glutamate into DAG under the action of PLCβ through GPPCR, and DAG promoted the synthesis of PC and PE under the action of PKCα and PLD, indicating that oat antimicrobial peptides inhibited the generation of inflammation by promoting the production of phospholipids.

#### 3.4.2. Trend of Gene Expression during Anti-Colitis Action of Antimicrobial Peptides

The colitis group and the AMP group DEGS were further analyzed using the clusterProfiler 4.0 for KEGG enrichment analysis. DEGS were mainly associated with colitis (Figure 5A), and colitis-related genes were subject to KEGG enrichment analysis again (Figure 5B). As can be seen from Figure 5A,B, the top four enrichment pathways were identified in relation to inflammatory bowel disease, Th17 cell differentiation, Th1 and Th2 cell differentiation, and wnt metabolic signaling pathways. It can be seen that these four pathways are the main metabolic pathways affecting colitis. However, the correlation between AMP and the four pathways is unclear, so further analysis is conducted by GSEA.

GSEA analysis was used to construct the gene regulatory network related to the prevention of colitis mediated by antimicrobial peptides. GSEA analyzed the relationship between gene expression and antimicrobial peptides in terms of colitis, Th17 cell differentiation, Th1 and Th2 cell differentiation, and wnt pathways. The oat antibacterial peptides down-regulated the gene set of the colitis pathway (NES = −2.366, Figure 6B) and also showed a negative correlation and inhibited the occurrence of colitis. Moreover, it inhibited the Th17 cell differentiation (NES = −2.808, Figure 6C), Th1 and Th2 cell differentiation (NES = −2.869, Figure 6A), and the production of inflammatory cells. Furthermore, the expression of the wnt pathway gene set was up-regulated (NES = 1.999, Figure 6D), which was also positively correlated and promoted cell proliferation and differentiation.

A total of 88 DEGs regulated by four pathways were screened out for protein–protein interaction (PPI) network analysis (Figure 7). The results showed that 25 of the genes were related to each other as Hub genes for the prevention of colitis via oat antimicrobial peptides.

#### 3.4.3. Western Blot Verification Experiment

The wnt (Wingless/Integrated) signaling pathway plays an important role in the self renewal, dryness, proliferation and differentiation of intestinal epithelial cells. DSS model can activate the Hippo signaling pathway and degrade YAP expression. At the same time, it can activate the wnt/β-catenin signaling pathway and inhibit the expression of the targeted factor β-catenin in the pathway. In 2018, Deng Feihong’s study found that YAP expression in UC was negatively correlated with the severity of intestinal inflammation, and the more severe the intestinal inflammation, the lower the YAP expression [25]. The expression of YAP in the nucleus can activate the expression of β-catenin in the wnt/β-catenin signaling pathway and the transcriptional regulation function. Meanwhile, YAP mediates the expression of proliferative indicators of intestinal epithelial cells by regulating wnt/β-catenin signaling, thus mediating the proliferation and repair of intestinal epithelial cells.

Using GAPDH as the internal reference protein, the relative protein expression level of lgr5, cyclin D1, β-catenin and YAP were measured in the colons of the M-AMP group and enteritis group. The expression level of apoptotic protein cyclin D1 was decreased, whereas those of lgr5, cyclin D1, and YAP were increased in the AMP group. However, only the apoptotic protein cyclin D1 expression was increased in the colitis group (Figure 8). Therefore, these results further verified that the oat antibacterial peptides could up-regulate the expression quantity of key genes in the wnt pathway, promote the lgr5 protein expression in the colon, and enhance the intestinal stem cell generation. It was proved that oat antimicrobial peptides can promote intestinal stem cell differentiation through the wnt signaling pathway.

#### 3.4.4. The Preventive Mechanism of Oat Antimicrobial Peptides on Colitis was Analyzed in Combination with Metabolome and Transcriptome

The potential regulatory gene network of oat antibacterial peptides on rat colitis was mapped by combining metabolome and transcriptome analysis (Figure 9). In this study, colitis occurred mainly because DSS over-activated the NF-Kb signaling pathway and promoted the up-regulation of the expression of inflammatory factor TNF-α in the pathway, leading to the occurrence of colitis. The major differential metabolite, diglycerol (DAG), can inhibit the TCR pathway and its downstream signaling, thereby inhibiting the production of inflammatory factors [26]. At the same time, the antimicrobial peptide activated the expression of the wnt frizzled gene in the wnt pathway, promoted the generation of intestinal stem cells, and promoted the differentiation and repair of intestinal epithelial cells.

## 4. Conclusions

DSS causes the glandular structure of the colon to disappear and be replaced by inflammatory tissue. It reduces the antioxidant capacity of colon cells, destroys the balance of the colon’s immune system, promotes the production of inflammatory factors, and inhibits the production of anti-inflammatory factors.

Through the analysis of metabolome and transcriptome, this experiment suggested that the main reason for enteritis is that DSS can over-activate the NF-ƙB signaling pathway, promote the up-regulation of the expression of inflammatory factor TNF-α in the pathway, and lead to the occurrence of enteritis. The balance between inflammatory T cells and regulatory T cells in the gut is the main mode for the body to regulate immune response [27]. The disorder of antigen presentation and Th1, Th2, Th7 and Treg of helper T cells activates specific host immune responses, induces immune response, and induces immune diseases such as IBD. Recent studies have found that IL-17 can promote the production of neutrophil chemokines in intestinal immunity, promote IBD response [28], and promote the secretion of pIgR (Ig receptor) and sIgA, which have anti-inflammatory effects [29]. IL-17 and IL-17R bind to inhibit effector cell Th1 cell-mediated IBD [30], and Th1 secretes cytokine IFN-ϒ to negatively regulate Th17 differentiation [31]. IL-17 has both pro-inflammatory and anti-inflammatory effects in IBD. However, it was found through metabolomics that oat antimicrobial peptides could promote DAG production to inhibit the activation of helper T cells, resulting in the down-regulation of key factors in the downstream major pathways Th1, Th2 and Th17 and the reduced expression of inflammatory factors. In 2023, DAG can detach T cell receptors (TCRS) from T cells, resulting in the termination of activation signals in T cells [32]. As a result, the T cell receptor pathway is inhibited, and further verification is needed. At the same time, the antimicrobial peptide activates the expression of the wnt frizzled gene in the wnt pathway, promotes the generation of intestinal stem cells, promotes the differentiation and repair of intestinal epithelial cells, and prevents colitis.

## Figures and Tables

**Figure 1 foods-13-00236-f001:**
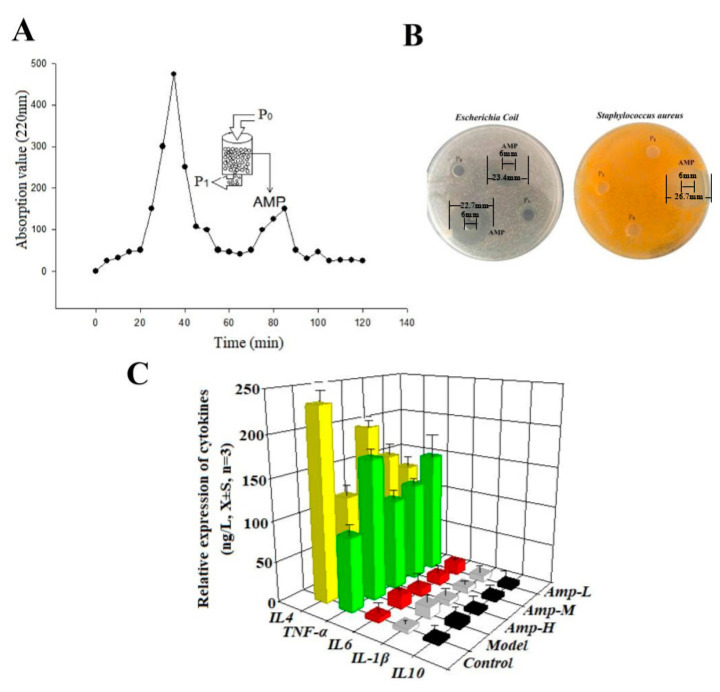
Enrichment of antimicrobial peptides and determination of their antibacterial and anti-inflammatory activities. (**A**) Chromatogram of cationic resin LSI-010. Oat protein hydrolyzed stock solution (P_0_), negatively charged and uncharged oat protein peptides not adsorbed by resin (P_1_), positively charged oat peptide (antibacterial active peptide, AMP). (**B**) Antibacterial experiment of oat antimicrobial peptides, AMP. (**C**) In Vitro anti-inflammatory experiment of oat antimicrobial peptides.

**Figure 2 foods-13-00236-f002:**
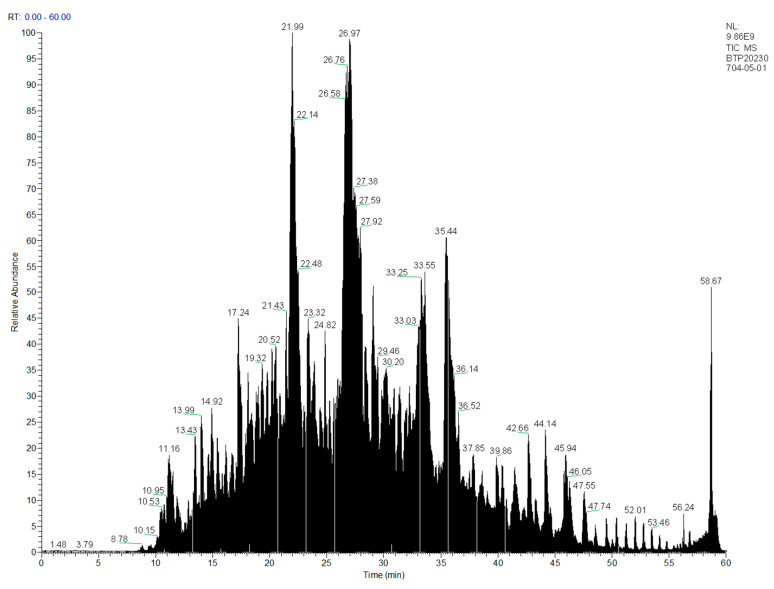
Sequence analysis of oat anti-polypeptide based on LC–MS/MS.

**Figure 3 foods-13-00236-f003:**
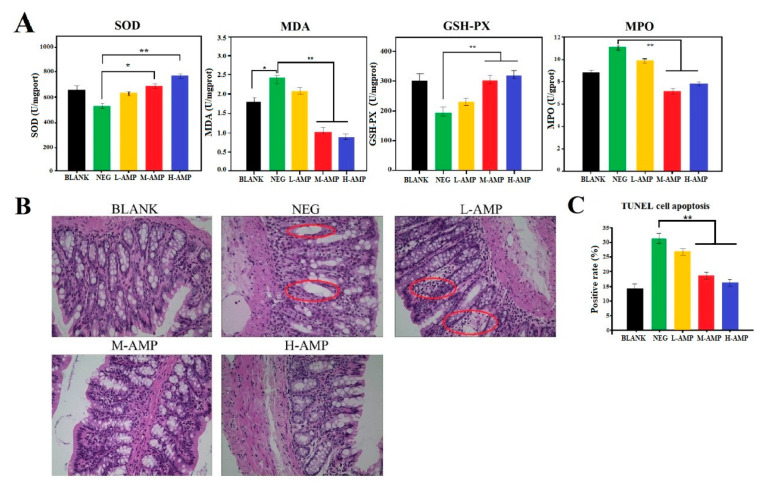
* means significant difference, ** means extremely significant difference. Effect of oat antimicrobial peptides on colitis in rats. (**A**) Determination of colon antioxidant capacity in rats. (**B**) Cell structure of rat colon. (**C**) Determination of apoptosis rate of colonic cells in rats.

**Figure 4 foods-13-00236-f004:**
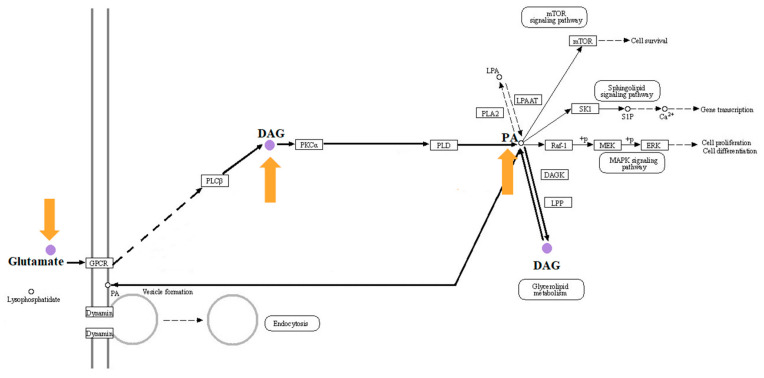
Phospholipid production metabolic pathway diagram [24].

**Figure 5 foods-13-00236-f005:**
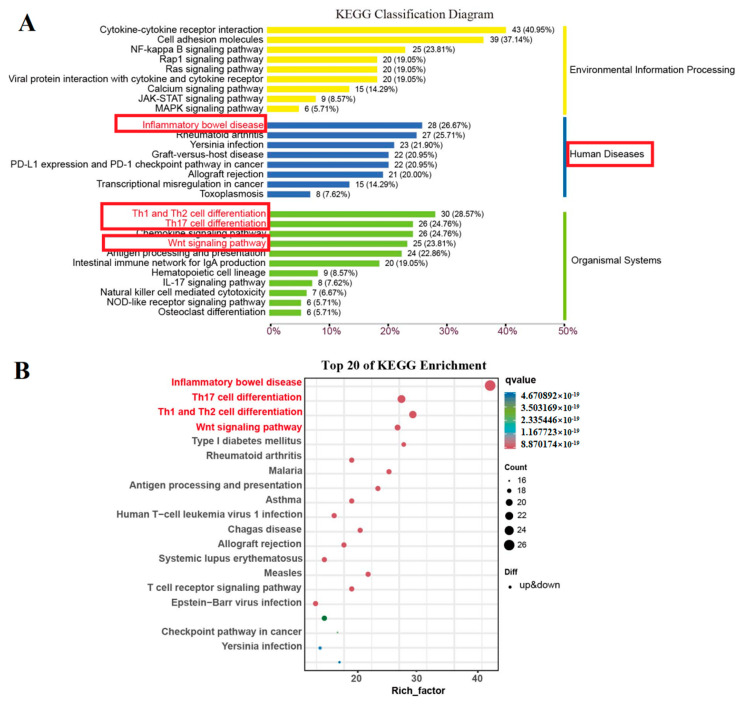
Differential metabolic pathway map of M-AMP and NEG.

**Figure 6 foods-13-00236-f006:**
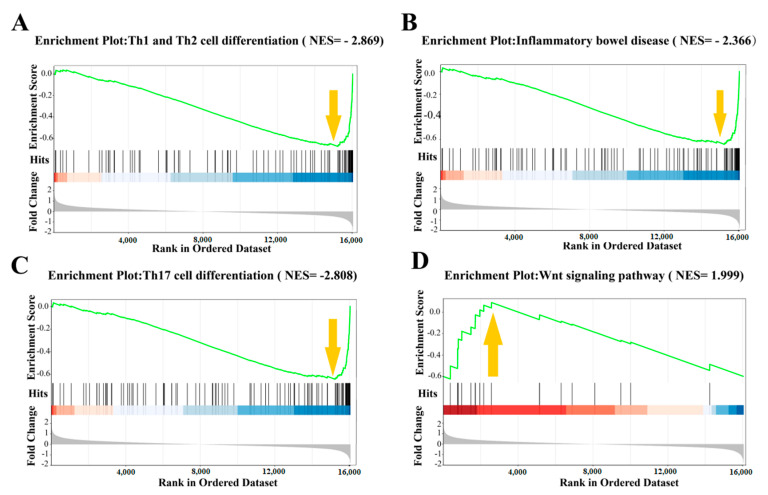
The downward arrow (↓) indicates that oat antimicrobial peptides down-regulate the gene set of this pathway, and the upward arrow (↑) indicates that oat antimicrobial peptides up-regulate the gene set of this pathway. Gene function analysis of oat antimicrobial peptides. (**A**–**D**) Results of gene set enrichment analysis on a representative gene set of oat antimicrobial peptide intervention.

**Figure 7 foods-13-00236-f007:**
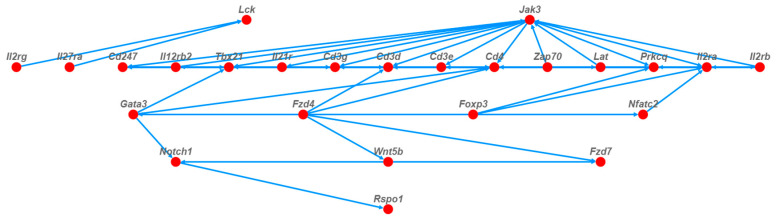
Protein–protein interaction networks of key gene proteins of oat antibacterial peptides for prevention of colitis.

**Figure 8 foods-13-00236-f008:**
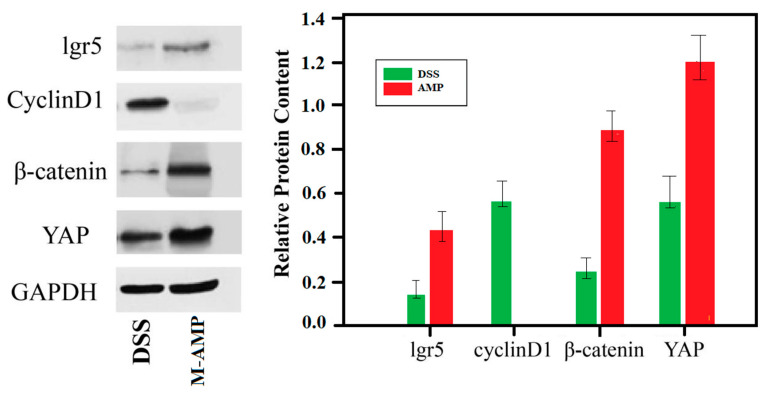
The key protein expression of the wnt signaling pathway in the colon.

**Figure 9 foods-13-00236-f009:**
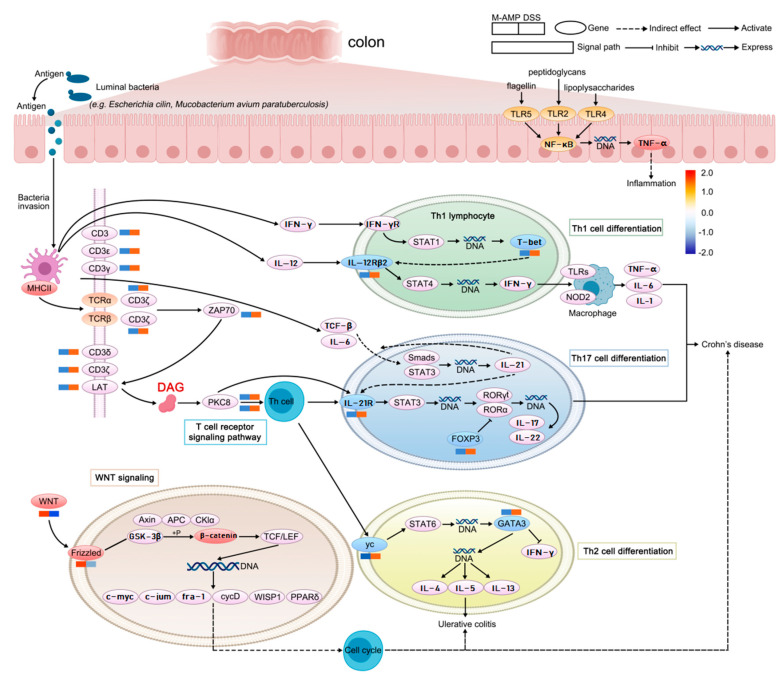
Potential regulatory gene network map of oat antimicrobial peptides on rat colitis.

**Table 1 foods-13-00236-t001:** Primary structure of oat antimicrobial peptides with scores > 500.

Peptide	z	Observed (M + H)	Calc.mass (M + H)	Mass Error (ppm)	Score
T.VIDAPGHRDFIKN.M	2	1481.795	1481.786	6.1	654.0
T.VIDAPGHRDF.I	2	1126.563	1126.564	−1.2	598.4
L.EDLRIPTAY.V	2	1077.556	1077.558	−1.0	588.0
K.IGGIGTVPVGRVE.T	2	1253.723	1253.721	1.0	579.3
V.IDAPGHRDF.I	2	1027.492	1027.496	−3.4	514.7

**Table 2 foods-13-00236-t002:** Major different metabolites in the NEG and AMP groups (|log2FC| > 1.5, *p*-value < 0.05, VIP > 1).

Metabolite Name	NEG	AMP	Regulated	Metabolite Name	NEG	AMP	Regulated
PC (18:0/18:4(6Z, 9Z, 15Z))	4189.29	65,640.01	up	YM-53601	18,941.12	15,287.58	down
PC (16:1(9Z)/20:3(5Z, 11Z))	3102.32	38,582.48	up	DL-Leucine	5665.91	4322.96	down
PC (14:0/22:4(7Z, 10Z, 16Z))	9856.89	112,375.47	up	3,5,5-Trimethyl 1,2-cyclohexanedione	118.79	84.13	down
Cysteinylglycine	14.49	75.22	up	10-Acetoxyligustroside	376.28	252.54	down
PE(15:0/22:1(13Z))	69,607.06	529,159.57	up	Indole	1058.07	679.63	down
PC (18:0/20:4(8Z, 11Z, 17Z))	3716.69	16,123.48	up	2-Aminomethylpyrimidine (hydrochloride)	959.22	615.64	down
PC (18:3(6Z, 12Z)/20:1(11Z))	17,535.9	70,552.72	up	Methionyl-Valine	106.73	68.24	down
PC (18:3(9Z, 12Z, 15Z)/20:0)	19,331.62	67,597.27	up	N-Malonyltryptophan	1906.65	1195.56	down
Glucosylceramide (18:1/18:0)	14,091.41	46,609.31	up	L-Histidine	1423.92	895.25	down
PC (18:0/22:5(7Z, 16Z, 19Z))	2056.87	6252.93	up	N-Undecanoylglycine	4919.8	3024.44	down
PS (17:1(9Z)/22:2(13Z, 16Z))	500.16	1435.46	up	Lysyl-Glutamine	307.4	187.89	down

## Data Availability

Data is contained within the article.

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
