# Peer review of "Potential Regulatory Gene Network Associated with the Ameliorative Effect of Oat Antibacterial Peptides on Rat Colitis"

_foods, 2024, doi:10.3390/foods13020236_

Round 1
Reviewer 1 Report
Comments and Suggestions for Authors
The manuscript describes the effect of oat antimicrobial peptides on the intestinal metabolite profile and antioxidant capacity in a rat colitis model. Moreover, high throughput multi-omics methods were used to analyze the effect and mechanism of oat antimicrobial peptides on colitis.
The study was well conducted and gathered a considerable amount of data. The results are interesting and relevant, especially by proposing a mechanism of action of oat AMPs on the intestine.
In general, the manuscript needs more discussion and comparison to other studies. The section “Results and Discussion” is limited to describing the results, with few in-depth discussion points.
Bellow are indicated some issues to be addressed:
INTRODUCTION:
Page 2, Line 58. “and enhancing,”. Enhancing what?
P2L58. “most of the related studies focus on physiology and ecology, and there are few reports on the molecular level.”. This phrase does not connect with the previous content (please separate them with a period). Physiology and ecology of what? The molecular level of what?
P2L69. “(…) proposed that theLgr5-labeled crypt base columnar (...)”. This phrase does not connect with the previous content (please separate them with a period). Who proposed it? Provide the names of the authors instead of numbers.
MATERIAL AND METHODS:
P3L123. Why did the authors use only trypsin to simulate the small intestine digestion phase? What about the other important proteases, such as chymotrypsin and elastase?
P3L137.”The absorption peak was obtained for the determination of antibacterial and anti-inflammatory activities”. You tested both peaks, right?
P3L142. What was the protein content of the samples submitted to the disc diffusion test?
P3L147. “ginger Salsa [10]”. Is that correct? The reference for this experiment did not perform the described method.
P4L155. Please describe the method accordingly. Do not use verbs in the imperative form.
P4L174. “Establishment of animal model of colitis and preventive effects of colitis assessment”. Did you develop this model? As far as I know, the use of DSS for inducing colitis was established before, right? You could clarify this section by not using the expression “establishment of a model”.
P4L174. What did the rats eat? Was the water/treatment consumption ad libitum, or was the volume controlled and administered by gavage?
P4L181. “10 mg/200 g”. Please explain the unit. 200g of what, body weight?
RESULTS AND DISCUSSION:
P7. Figure 1. Please increase the size of antimicrobial activity assay photos. Provide a descriptive caption for sample names.
P7L312. As I understood, the method did not simulate oral chewing since it did not replicate the shear movement of the teeth.
P7L318. “Peptides”, not “peptide”. Please provide a discussion regarding the disc diffusion test. Did you measure the inhibition halo? Compare your results to other studied AMPs.
P7L319-325 Please rewrite the text, emphasizing which samples achieved statistically significant differences from the model. Moreover, IL-6 is repeated in the text, being mentioned as increased and decreased in the experiment.
P7L324. “(…) indicating that AMP had good anti-inflammatory activity.” All concentrations performed equally? Were the results statistically comparable to the Control?
P8L332. “It is better able to bind to cell membranes and act on inflammatory cells.” Rewrite to improve clarity.
P8L336. Provide a clear title for this figure. It does not bring the primary structure of the peptide, does it?
Figure 3. Please provide a descriptive caption for sample names and stats results (* and **).
Section 3.3. Please provide a discussion regarding these results. Compare your results to other studied AMPs or classical medicines used to treat colitis.
P10L375. The role of phospholipids PC and PE on intestinal metabolism should be discussed.
Figure 5. Provide a proper title. It must describe what is presented in the figure rather than what it concludes.
Figure 8. Which of the 3 AMP groups was used in this analysis? Please provide this information in the caption. The figure must be readable alone.
Section 3.2.3. Please provide a discussion regarding these results. Compare your results to other studies.
CONCLUSION:
P14L463-465. This is not a conclusion of your work.
P14L469. There is no need to mention Fig 3 in the conclusion section.
P14L478-479. “In 2023, the exocytosis of immune synapses down-regulates the TCR pathway”. Please rewrite for clarity.
Comments on the Quality of English LanguageSome parts of the manuscript need attention to improve clarity.
Author Response
For research article
Response to Reviewer 1 Comments Thank you very much for taking the time to review this manuscript. According to your valuable comments and suggestions, we have revised the paper and marked it in red. Comments 1: 1 Comments and Suggestions for Authors: The manuscript describes the effect of oat antimicrobial peptides on the intestinal metabolite profile and antioxidant capacity in a rat colitis model. Moreover, high throughput multi-omics methods were used to analyze the effect and mechanism of oat antimicrobial peptides on colitis. The study was well conducted and gathered a considerable amount of data. The results are interesting and relevant, especially by proposing a mechanism of action of oat AMPs on the intestine. In general, the manuscript needs more discussion and comparison to other studies. The section “Results and Discussion” is limited to describing the results, with few in-depth discussion points.
2 Bellow are indicated some issues to be addressed: INTRODUCTION: (1) Page 2, Line 58. “and enhancing,”. Enhancing what? (2) P2L58. “most of the related studies focus on physiology and ecology, and there are few reports on the molecular level.”. This phrase does not connect with the previous content (please separate them with a period). Physiology and ecology of what? The molecular level of what? (3) P2L69. “(…) proposed that theLgr5-labeled crypt base columnar (...)”. This phrase does not connect with the previous content (please separate them with a period). Who proposed it? Provide the names of the authors instead of numbers. MATERIAL AND METHODS: (4) P3L123. Why did the authors use only trypsin to simulate the small intestine digestion phase? What about the other important proteases, such as chymotrypsin and elastase? (5) P3L137.”The absorption peak was obtained for the determination of antibacterial and anti-inflammatory activities”. You tested both peaks, right? (6) P3L142. What was the protein content of the samples submitted to the disc diffusion test? (7) P3L147. “ginger Salsa [10]”. Is that correct? The reference for this experiment did not perform the described method. (8) P4L155. Please describe the method accordingly. Do not use verbs in the imperative form. (9) P4L174. “Establishment of animal model of colitis and preventive effects of colitis assessment”. Did you develop this model? As far as I know, the use of DSS for inducing colitis was established before, right? You could clarify this section by not using the expression “establishment of a model”. (10) P4L174. What did the rats eat? Was the water/treatment consumption ad libitum, or was the volume controlled and administered by gavage? (11) P4L181. “10 mg/200 g”. Please explain the unit. 200g of what, body weight? RESULTS AND DISCUSSION: (12) P7. Figure 1. Please increase the size of antimicrobial activity assay photos. Provide a descriptive caption for sample names. (13) P7L312. As I understood, the method did not simulate oral chewing since it did not replicate the shear movement of the teeth. (14) P7L318. “Peptides”, not “peptide”. Please provide a discussion regarding the disc diffusion test. Did you measure the inhibition halo? Compare your results to other studied AMPs. (15) P7L319-325 Please rewrite the text, emphasizing which samples achieved statistically significant differences from the model. Moreover, IL-6 is repeated in the text, being mentioned as increased and decreased in the experiment. (16) P7L324. “(…) indicating that AMP had good anti-inflammatory activity.” All concentrations performed equally? Were the results statistically comparable to the Control? (17) P8L332. “It is better able to bind to cell membranes and act on inflammatory cells.” Rewrite to improve clarity. (18) P8L336. Provide a clear title for this figure. It does not bring the primary structure of the peptide, does it? (19) Figure 3. Please provide a descriptive caption for sample names and stats results (* and **). (20) Section 3.3. Please provide a discussion regarding these results. Compare your results to other studied AMPs or classical medicines used to treat colitis. (21) P10L375. The role of phospholipids PC and PE on intestinal metabolism should be iscussed. (22) Figure 5. Provide a proper title. It must describe what is presented in the figure rather than what it concludes. (23) Figure 8. Which of the 3 AMP groups was used in this analysis? Please provide this information in the caption. The figure must be readable alone. (24) Section 3.2.3. Please provide a discussion regarding these results. Compare your results to other studies. CONCLUSION: (25) P14L463-465. This is not a conclusion of your work. (26) P14L469. There is no need to mention Fig 3 in the conclusion section. (27) P14L478-479. “In 2023, the exocytosis of immune synapses down-regulates the TCR pathway”. Please rewrite for clarity. 3 Comments on the Quality of English Language Some parts of the manuscript need attention to improve clarity.
Question 1 : Page 2, Line 58. “and enhancing,”. Enhancing what? |
Response 1: We are very sorry for our negligence. We had modified it. Please see it in page 2, line 58-66. We have changed it. “For example, Regulate the tendency of immune cells and the differentiation of lymphocytes, and promote the production of lymphocytes, cytokines and tendency factors [4,5]. Antimicrobial peptide LL-37 can regulate the apoptosis and activation of macrophages and neutral cells [6], promote the clearance of host pathogens and enhance anti-infection ability. Chronic inflammation leads to continuous bacterial stimulation of the host, resulting in tissue and organ damage. ” Question 2 : “most of the related studies focus on physiology and ecology, and there are few reports on the molecular level.”. This phrase does not connect with the previous content (please separate them with a period). Physiology and ecology of what? The molecular level of what? Response 2: We are very sorry for our negligence. We had modified it. Please see it in page 2, line 66-69. We have changed it. “Antimicrobial peptides have antibacterial and host immune regulation effects, and antimicrobial peptides have gradually attracted attention in the treatment of chronic inflammation [7-8].” Question 3 : P2L69. “(…) proposed that theLgr5-labeled crypt base columnar (...)”. This phrase does not connect with the previous content (please separate them with a period). Who proposed it? Provide the names of the authors instead of numbers. Response 3: We are very sorry for our negligence. We had been deleted. Question 4 : P3L123. Why did the authors use only trypsin to simulate the small intestine digestion phase? What about the other important proteases, such as chymotrypsin and elastase? Response 4: This method is referred to in reference 14, which is an internationally recognized method of in vitro digestion. It is presumed that trypsin can activate chymotrypsin activity, so only trypsin's hydrolytic activity on proteins is considered here The experiment was carried out according to static in vitro simulation INFOGEST method, Question 5 : P3L137.”The absorption peak was obtained for the determination of antibacterial and anti-inflammatory activities”. You tested both peaks, right? Response 5: Yes, Professor. in vitro validation trials, we have measured their anti-inflammatory activity Question 6 : P3L142. What was the protein content of the samples submitted to the disc diffusion test? Response 6: Tests conducted directly with purified freeze-dried powder Question 7: P3L147. “ginger Salsa [10]”. Is that correct? The reference for this experiment did not perform the described method. Response 7: We are very sorry for our negligence. We had been changed it. Please see it in page 4, line 169. Question 8: P4L155. Please describe the method accordingly. Do not use verbs in the imperative form. Response 8: We are very sorry for our negligence. We had been changed it. Please see it in page 4, line 152-165. Question 9: P4L174. “Establishment of animal model of colitis and preventive effects of colitis assessment”. Did you develop this model? As far as I know, the use of DSS for inducing colitis was established before, right? You could clarify this section by not using the expression “establishment of a model”. Response 9: We are very sorry for our negligence. We had been changed it. Please see it in page 4, line 180. Question 10: P4L174. What did the rats eat? Was the water/treatment consumption ad libitum, or was the volume controlled and administered by gavage? Response 10: The food is ordinary rat food, DSS was dissolved in ordinary drinking water, the water was ad libitum, but antimicrobial peptides were fed by gavage. Question 11: P4L181. “10 mg/200 g”. Please explain the unit. 200g of what, body weight? Response 11: 200g is the weight of a rat, We are very sorry for our negligence. We had been changed it. Please see it in page 4, line 185-189. Question 12: P7. Figure 1. Please increase the size of antimicrobial activity assay photos. Provide a descriptive caption for sample names. Response 12: We are very sorry for our negligence. We had been changed it. Please see it in page 6, line 236. Question 13: P7L312. As I understood, the method did not simulate oral chewing since it did not replicate the shear movement of the teeth. Response 13: we planted to simulate human digestion in vitro, so we added oral digestion Question 14: P7L318. “Peptides”, not “peptide”. Please provide a discussion regarding the disc diffusion test. Did you measure the inhibition halo? Compare your results to other studied AMPs. Response 14: We are very sorry for our negligence. We had been changed it. Please see it in page 6, line 253-264. Question 15: P7L319-325 Please rewrite the text, emphasizing which samples achieved statistically significant differences from the model. Moreover, IL-6 is repeated in the text, being mentioned as increased and decreased in the experiment. Response 15: We are very sorry for our negligence. We had been changed it. Please see it in page 7, line 265-268. Question 16: P7L324. “(…) indicating that AMP had good anti-inflammatory activity.” All concentrations performed equally? Were the results statistically comparable to the Control? Response 16: We are very sorry for our negligence. We had been changed it. Please see it in page 7, line 269-272. Question 17: P8L332. “It is better able to bind to cell membranes and act on inflammatory cells.” Rewrite to improve clarity. Response 17: We are very sorry for our negligence. We had been changed it. Please see it in page 7, line 281-285. Question 18: P8L336. Provide a clear title for this figure. It does not bring the primary structure of the peptide, does it? Response 18: We are very sorry for our negligence. We had been changed it. Please see it in page 7, line287. Question 19: Figure 3. Please provide a descriptive caption for sample names and stats results (* and **). Response 19: We are very sorry for our negligence. We had been changed it. Please see it in page 8, line302-304. Question 20: Section 3.3. Please provide a discussion regarding these results. Compare your results to other studied AMPs or classical medicines used to treat colitis. Response 20: We are very sorry for our negligence. We had been changed it. Please see it in page 9, line319-324. Question 21: P10L375. The role of phospholipids PC and PE on intestinal metabolism should be iscussed. Response 21: As the professor said, lipids are conducive to improving the immune capacity of the body, which needs to be confirmed. This conclusion is based on the experiment, different metabolites are screened for different experimental groups, and relevant metabolic pathways are identified, which is a speculation that needs further verification. Question 22: Figure 5. Provide a proper title. It must describe what is presented in the figure rather than what it concludes. Response 22: We are very sorry for our negligence. We had been changed it. Please see it in page 10, line349. Question 23: Figure 8. Which of the 3 AMP groups was used in this analysis? Please provide this information in the caption. The figure must be readable alone. Response 23: We are very sorry for our negligence. We had been changed it. Please see it in page 12, line387. Question 24: Section 3.2.3. Please provide a discussion regarding these results. Compare your results to other studies. Response 24: We are very sorry for our negligence. We had been changed it. Please see it in page 12, line390-405. Question 25: P14L463-465. This is not a conclusion of your work. Response 25: We are very sorry for our negligence. We had been changed it. Please see it in page 14, line442-446. Question 26: P14L469. There is no need to mention Fig 3 in the conclusion section. Response 26: We are very sorry for our negligence. We had been deleted it. Question 27: P14L478-479. “In 2023, the exocytosis of immune synapses down-regulates the TCR pathway”. Please rewrite for clarity. Response 27: We are very sorry for our negligence. We had been changed it. Please see it in page 15, line468-470. |
Question 28:Comments on the Quality of English Language
Some parts of the manuscript need attention to improve clarity.
Response 28: Dear professor, this article is looking for the company to polish the language again, the certificate is as follows.

Reviewer 2 Report
Comments and Suggestions for Authors
The authors carried out a comprehensive study on the anti-inflammatory properties exhibited by peptides derived from oats. Interesting results were obtained. However, some comments should be addressed.
Line 29: express the abbreviation DSS
Line 58: enhancing? Please complet
Line 99: scientific names, italics. review the entire document
Line 112: What would be the purpose of oral digestion?
Line 126: Enzymes such as chymotrypsin, elastase... were not included. These proteases cleave peptide bonds made up of different residues, resulting in another pool of AMP. It would not be comparable with those obtained in the digestion process
Lines 140-142: Details such as concentration of the inoculum and others are necessary
Line 147: ginger salsa?; cell concentration?
Line 149: How was the peptide concentration obtained? Was it lyophilized and then resuspended? or it was quantified and then diluted. How was it sterilized?
Line 151: What does each group correspond to?
Line 155: this should be before 2.3
Line 181: Were the doses daily?
Line 200; What was it diluted with?
Line 210: the product is the precipitated?
Line 214: What are the quality control criteria?
Fig 1A: In the methodology it is written wavelength 280 nm
Line 328: If the cut-off was 5000Da, what happened to those with weights 1501-5000Da?
Line 351: It must be contextualized because some variables increase and decrease in accordance with the antioxidant capacity.
Line 360: Discuss with the results obtained in Fig 1
Lines 378-382: Here it must be better contextualized. According to Figure 4, glutamate is the ligand of the GPCR. Is this first messenger supposed to be provided by the peptide? clarify. If so, in Table 1, only the sequence in position 4 has 1 Glut. What function would the other peptides have? since those with a negative charge were excluded by ion exchange cormatography. Now, the activation of phospholipase c leads to the release of arachidonic acid, a precursor of prostaglandins, inflammatory substances, contradicting the evidence found. How is this addressed? discuss please (https://doi.org/10.1016/S0090-6980(97)00011-7)
Line 384: Reference should be included since the results obtained cannot describe such a signaling pathway.
Line 452: Reference should be included
Author Response
For research article
Response to Reviewer 2 Comments Thank you very much for taking the time to review this manuscript. According to your valuable comments and suggestions, we have revised the paper and marked it in red. Comments 2: Comments and Suggestions for Authors The authors carried out a comprehensive study on the anti-inflammatory properties exhibited by peptides derived from oats. Interesting results were obtained. However, some comments should be addressed. (1) Line 29: express the abbreviation DSS (2) Line 58: enhancing? Please complet (3) Line 99: scientific names, italics. review the entire document (4) Line 112: What would be the purpose of oral digestion? (5) Line 126: Enzymes such as chymotrypsin, elastase... were not included. These proteases cleave peptide bonds made up of different residues, resulting in another pool of AMP. It would not be comparable with those obtained in the digestion process (6) Lines 140-142: Details such as concentration of the inoculum and others are necessary (7) Line 147: ginger salsa?; cell concentration? (8) Line 149: How was the peptide concentration obtained? Was it lyophilized and then resuspended? or it was quantified and then diluted. How was it sterilized? (9) Line 151: What does each group correspond to? (10) Line 155: this should be before 2.3 (11) Line 181: Were the doses daily? (12) Line 200; What was it diluted with? (13) Line 210: the product is the precipitated? (14) Line 214: What are the quality control criteria? (15) Fig 1A: In the methodology it is written wavelength 280 nm (16) Line 328: If the cut-off was 5000Da, what happened to those with weights 1501-5000Da? (17) Line 351: It must be contextualized because some variables increase and decrease in accordance with the antioxidant capacity. (18) Line 360: Discuss with the results obtained in Fig 1 (19) Lines 378-382: Here it must be better contextualized. According to Figure 4, glutamate is the ligand of the GPCR. Is this first messenger supposed to be provided by the peptide? clarify. If so, in Table 1, only the sequence in position 4 has 1 Glut. What function would the other peptides have? since those with a negative charge were excluded by ion exchange cormatography. Now, the activation of phospholipase c leads to the release of arachidonic acid, a precursor of prostaglandins, inflammatory substances, contradicting the evidence found. How is this addressed? discuss please (https://doi.org/10.1016/S0090-6980(97)00011-7) (20) Line 384: Reference should be included since the results obtained cannot describe such a signaling pathway. (21) Line 452: Reference should be included
|
Question 1 : Line 29: express the abbreviation DSS Response 1: We are very sorry for our negligence. We have changed it. Please see it in page 1, line 24. Question 2 : Line 58: enhancing? Please complet Response 2: We are very sorry for our negligence. We have changed it. Please see it in page 2, line 58-69. Question 3 : Line 99: scientific names, italics. review the entire document Response 3: We are very sorry for our negligence. We have changed it. Please see it in page 3, line 104-105,148, page 6, line 262 Question 4 : Line 112: What would be the purpose of oral digestion? Response 4: Simulating the shear movement of human teeth Question 5 : Line 126: Enzymes such as chymotrypsin, elastase... were not included. These proteases cleave peptide bonds made up of different residues, resulting in another pool of AMP. It would not be comparable with those obtained in the digestion process Response 5: This method is referred to in reference 14, which is an internationally recognized method of in vitro digestion. As the professor said, we later carried out animal experiments to verify whether digestion in vivo affected its activity Question 6 : Lines 140-142: Details such as concentration of the inoculum and others are necessary Response 6: We are very sorry for our negligence. We had been changed it. Please see it in page 3, line 149. Question 7: Line 147: ginger salsa?; cell concentration? Response 7: We are very sorry for our negligence. We had been changed it. Please see it in page 4, line 170. Question 8: Line 149: How was the peptide concentration obtained? Was it lyophilized and then resuspended? or it was quantified and then diluted. How was it sterilized? Response 8: The graded peptide solution was lyophilized and then filtered to remove bacteria Question 9: Line 151: What does each group correspond to? Response 9: We are very sorry for our negligence. We had been changed it. Please see it in page 4, line 169-172. Question 10: Line 155: this should be before 2.3 Response 10: Ok, We had been changed it. Question 11: Line 181: Were the doses daily? Response 11: Yes, we dosed the rats every day Question 12: Line 200; What was it diluted with? Response 12: We dilute it with pre-cooled brine. Question 13: Line 210: the product is the precipitated? Response 13: The product is supernatant. Question 14: Line 214: What are the quality control criteria? Response 14: By deleting it from the original data sequences that contain joints, sequences that contain ploy-N, and sequences of low quality to obtain Clean data Question 15: Fig 1A: In the methodology it is written wavelength 280 nm Response 15: The UV detection wavelength of small molecule peptides is mostly 220nm, and our experiment found that the detection value of 220nm is higher than that of 280nm Question 16: Line 328: If the cut-off was 5000Da, what happened to those with weights 1501-5000Da? Response 16: We graded the peptides after chromatographic separation and found that the peptides below 5000 Dalton had the strongest activity. Question 17: Line 351: It must be contextualized because some variables increase and decrease in accordance with the antioxidant capacity. Response 17: We are very sorry for our negligence. We had been changed it. Please see it in page 8, line302-304. Question 18: Line 360: Discuss with the results obtained in Fig 1 Response 18: We are very sorry for our negligence. We had been changed it. Please see it in page6, line319-324. Question 19: Lines 378-382: Here it must be better contextualized. According to Figure 4, glutamate is the ligand of the GPCR. Is this first messenger supposed to be provided by the peptide? clarify. If so, in Table 1, only the sequence in position 4 has 1 Glut. What function would the other peptides have? since those with a negative charge were excluded by ion exchange cormatography. Now, the activation of phospholipase c leads to the release of arachidonic acid, a precursor of prostaglandins, inflammatory substances, contradicting the evidence found. How is this addressed? discuss please (https://doi.org/10.1016/S0090-6980(97)00011-7) Response 19: Sodium glutamate may not be the key substance promoting diglycerol, the conclusion in the paper is a speculative result, still need to be further verified. Question 20: Line 384: Reference should be included since the results obtained cannot describe such a signaling pathway. Response 20: We are very sorry for our negligence. We had been changed it. Please see it in page 10, line345. Question 21: Line 452: Reference should be included Response 21: We are very sorry for our negligence. We had been changed it. Please see it in page 13, line 432.
|

Reviewer 3 Report
Comments and Suggestions for Authors
The research findings on the mechanism associated with colitis using antimicrobial peptides are expected to provide valuable academic insights for researchers in this field. Additionally, the utilization of antimicrobial peptides derived from corn is perceived to have a positive impact on the application of plant-based peptides. The notable aspect is the confirmation that antimicrobial peptides inhibit the activation of T helper cells, downregulate key factors in TH1, TH2, and TH17 pathways, and suppress the generation of inflammatory cells. This is noteworthy for its potential implications in the field.
However, it is advisable to provide a detailed conclusion with references related to the T cell receptor pathway to validate the aspects requiring further verification. This would enhance the credibility of the findings.
Author Response
For research article
Response to Reviewer 3 Comments Thank you very much for taking the time to review this manuscript. According to your valuable comments and suggestions, we have revised the paper and marked it in red. Comments 3: Comments and Suggestions for Authors The research findings on the mechanism associated with colitis using antimicrobial peptides are expected to provide valuable academic insights for researchers in this field. Additionally, the utilization of antimicrobial peptides derived from corn is perceived to have a positive impact on the application of plant-based peptides. The notable aspect is the confirmation that antimicrobial peptides inhibit the activation of T helper cells, downregulate key factors in TH1, TH2, and TH17 pathways, and suppress the generation of inflammatory cells. This is noteworthy for its potential implications in the field. However, it is advisable to provide a detailed conclusion with references related to the T cell receptor pathway to validate the aspects requiring further verification. This would enhance the credibility of the findings.
Response We are very sorry for our negligence. We have changed and marked it in red.. Please see conclusion.
|
